# Differential expression of heat shock proteins and antioxidant enzymes in response to temperature, starvation, and parasitism in the Carob moth larvae, *Ectomyelois ceratoniae* (Lepidoptera: Pyralidae)

Saeed Farahani[1], Ali R. Bandani[1]*, Houshang Alizadeh[2], Seyed Hossein Goldansaz[1], Steven Whyard[3]

**1** Plant Protection Department, College of Agriculture and Natural Resources, University of Tehran, Karaj, Iran, **2** Department of Agronomy and Plant Breeding, College of Agriculture and Natural Resources, University of Tehran, Karaj, Iran, **3** Department of Biological Sciences, University of Manitoba, Winnipeg, Manitoba, Canada

* abandani@ut.ac.ir

## Abstract

Insects face diverse biotic and abiotic stresses that can affect their survival. Many of these stressors impact cellular metabolism, often resulting in increased accumulation of reactive oxygen species (ROS). Consequently, insects will respond to these stressors by increasing antioxidant activity and increased production of heat shock proteins (HSPs). In this study, the effect of heat, cold, starvation, and parasitism by *Habroacon hebetor* wasps was examined in the carob moth, *Ectomyelois ceratoniae*, to determine which responses were common to different stresses. For all stressors, malondialdehyde levels increased, indicative of oxidative stress in the insects. The activity of two antioxidant enzymes, superoxide dismutase (SOD) and catalase (CAT), increased with each stress, suggesting that these enzymes were serving a protective role for the insects. Heat (46˚C for 100 min) and cold (-15˚C for 30 min) treatments caused significant mortalities to all developmental stages, but pretreatments of moderate heat (37˚C for 10 min) or cold (10˚C for 10 min) induced thermotolerance and reduced the mortality rates when insects were subsequently exposed to lethal temperatures. Quantitative RT-PCR confirmed that heat and cold tolerance were associated with up-regulation of two HSPs, HSP70 and HSP90. Interestingly, HSP70 transcripts increased to a greater extent with cold treatment, while HSP90 transcripts increased more in response to high temperatures. RNA interference (RNAi)-mediated knockdown of either HSP70 or HSP90 transcripts was achieved by injecting larvae with dsRNA targeting each gene's transcripts, and resulted in a loss of acquired thermotolerance in insects subjected to the heat or cold pretreatments. These observations provide convincing evidence that both HSP70 and HSP90 are important mediators of the acquired thermotolerance. Starvation and parasitism by wasps caused differential expression of the HSP genes. In response to starvation, HSP90 transcripts increased to a greater extent than HSP70, while in contrast, HSP70 transcripts increased to a greater extent than those of HSP90 during the first 48 h of wasp

**Data Availability Statement:** All relevant data are within the manuscript and its Supporting Information files.

**Funding:** This research was supported by funds to Ali R. Bandani from the University of Tehran, funds to Steven Whyard from Natural Sciences and Engineering Research Council, and a scholarship from Iran's Minister of Science Research and Technology to Saeed Farahani. The funder had no role in study design, data collection and analysis, decision to publish, or preparation of the manuscript.

**Competing interests:** The authors have declared that no competing interests exist.

parasitism. These results showed the differential induction of the two HSPs' transcripts with variable stresses. As well as, heat, cold, starvation, and parasitism induce oxidative stress, and antioxidant enzymes likely play an important role in reducing oxidative damage in *E. ceratoniae*.

## Introduction

Insects face a variety of biotic (pathogens, predators, and parasitoids) and abiotic (high/low temperature, UV radiation, ultrasound waves, chemicals, etc.) external stressors [1,2]. Being aerobes, insects also must deal with endogenous sources of chemical stress, as they continuously produce reactive oxygen species (ROS), which form during the reduction of $O_2$ [2]. ROS compounds can damage cellular lipids, proteins, and nucleic acids [2], and hence, insects, like other organisms, have developed protective responses to detoxify ROS compounds before damage occurs.

The antioxidant system in most aerobes consists of both enzymes and non-enzymatic low-molecular-weight antioxidants, both of which contribute to reducing the levels of ROS during stress conditions [3]. In insects, the most important antioxidant enzymes are superoxide dismutase (SOD), peroxidase (POX), catalase (CAT) and glutathione S-transferase (GST) [2]. SOD catalyzes superoxide radicals ($O_2^-$) into oxygen ($O_2$) and hydrogen peroxide ($H_2O_2$), and the $H_2O_2$ is then converted by both CAT and POX into oxygen and water ($H_2O$) [2; 3]. In addition to the antioxidant enzymes, there are non-enzymatic compounds that play a role in protecting insects against ROS compounds, including lipid-soluble and water-soluble antioxidants [2]. The antioxidant system is well-regulated, controlling ROS levels during normal metabolism, but also capable of increasing enzyme activities and antioxidants during various stresses. Induction of the antioxidant system has been observed in many insects subjected to different stresses, such temperature [2; 4; 5; 6], dietary oxidants [7], pesticide exposures [8; 9], oxygen deprivation [10] and infections or parasitisms [11; 12].

In addition to the antioxidant stress system, heat shock proteins can also provide a protective function to stressed cells. While they are named heat shock proteins, they have been found to be produced in response to various biotic stresses (parasites, pathogens, and parasitoids) and abiotic stresses (temperature, ultraviolet radiation, drought dehydration and anhydrobiosis, chemicals and metals [1; 13]. In insects, four major HSP families have been recognized, including the small heat shock proteins (smHSPs), Hsp60, Hsp70, and Hsp90 [14]. These proteins serve a variety of functions within cells, but are particularly important in serving as molecular chaperones, helping nascent proteins fold correctly, or helping proteins refold if they partially denature during normal cellular activities. Due to their protein chaperone functions, HSPs play important roles in regulating growth, development, and diapause, and participate in many physiological processes including oogenesis, embryo development, and signal transduction in insects [14; 15; 16; 17]. During many cellular stresses, proteins can denature, and HSPs then provide a protection function, enabling proteins to refold into their native form. Thus, HSPs act as a first line of the cellular defense by preventing irreversible denaturation of essential proteins under biotic and abiotic stress conditions [18; 6]. There are numerous studies illustrating the role of HSPs in insects (reviewed in 1). The vast majority of these studies provide evidence of increased transcription of different HSPs during heat stress, but cold stress is also a common inducer of HSP transcription [1]. Other stresses have also been observed to induce HSP gene transcription. For example, Yocum et al. [19] found that expression of the Hsp23 transcript of the nondiapausing flesh fly was induced in response to both

severe heat and cold shocks (43ºC and -10ºC for 2 h). Sinclair et al. [20] reported that cold exposure induced HSP70 expression in *Drosophila melanogaster*. Rinehart et al. [21] showed that in the pupae of the flesh fly, *Sarcophaga crassipalpis*, HSP23 and HSP70 levels were significantly increased after envenomation by the endoparasitoid, *Nasonia vitripennis*. Wang et al. [22] observed that the expression of HSPs was significantly higher in the migratory phase of *Locusta migratoria* L. than the solitary phase, suggesting that crowding mediates HSP gene activation. Shim et al. [23] reported that the expression levels of smHSP and HSC70 in larvae of *Plodia interpunctella* increased in response to *H. hebetor* parasitism. Wang et al. [24] showed the HSP60, HSP70, and HSP90 expression levels increased after 24 h starvation stress in *Pteromalus puparum*.

The ability of organisms to survive exposure to high/low temperature levels is defined as heat/cold hardiness and its induction is known as acclimation [25–26]. Acclimation is associated with the production of HSPs, antioxidant enzymes/compounds, polyols and sugars. Insects can alter their sensitivity to heat/cold stress through short-term acclimation or long-term evolutionary adaptation [27]. Short-term heat/cold acclimation can involve physiological and biochemical changes in insects, enabling them to adapt to variable environmental temperatures [28–30]. Acclimation for a few days at low temperatures generally improves cold hardiness (reviewd in:[31]). Likewise, rapid cold hardening with acclimation for 2 h at 4ºC of recently-eclosed adults of five coleopteran species associated with stored grain significantly increased the survival time at various sub-zero temperature compared to those without acclimation [32]. Yocum and Denlinger [19] likewise showed that a mild heat treatment of 40˚C for 2 h to the flesh fly, *Sarcophaga crassipalpis*, conferred thermal tolerance to a subsequent normally lethal heat treatment of 90 min at 45˚C.

The carob moth, *Ectomyelois ceratoniae* (Lep. Pyralidae), is a highly destructive polyphagous pest insect of fruit, nut crops and stored products, with feeding larvae causing considerable damage in orchards and food stores. Typically, it is controlled through the use of synthetic pesticides such as methyl bromide and phosphine [33], but with increasing concerns about the negative impacts of these chemicals on the environment, alternative methods, such as extreme temperature treatments or other inducers of oxidative stress are worthy of more consideration. As little is known of this insect's ability to withstand temperature extremes, this study examined the effect of heat and cold stresses on the insect's antioxidant responses and changes in expression of two HSP (HSP70 and HSP 90) genes. Other, non-chemical methods of control of Pyralid moth have been investigated, including the use of parasitoid wasps [34]. To explore whether biotic stresses might induce similar stress responses in the carob moths, we also examined whether parasitism by *Habroacon hebetor* induced similar or different stress responses to those seen following temperature stress. Another common stress that insects may encounter is brief periods of starvation, particularly in environments where food availability can change quickly, such as during harvest periods or food shipments. As starvation has been found to induce some common cellular stress responses in other insects [14, 20, 35], we also examined whether brief starvation elicited any of the responses observed with the other stresses, to determine whether some responses were shared across seemingly different types of stress. By understanding how this pest insect responds to different stresses, we may be able to design more effective methods of control that counter their natural protective or curative stress responses.

## Materials and methods

### Insect cultures

Carob moth larvae-infested pomegranate fruits were collected from an orchard in Tehran Province, Iran (Latitude: 35.282384 Longitude: 51.750282). The fruit, containing the larvae,

were transferred to screened cages kept within a growth chamber (26±1˚C, 70% humidity, and 16:8 h L: D). After adult moth emergence, they were collected by an aspirator and 70 adult moths (mixed sex) were transferred to egg-laying containers. The four-liters cylindrical plastic containers were lined with a fine cotton cloth as a suitable surface on which to lay their eggs. After three days, the eggs were collected and were transferred to artificial diets. The artificial larval diet was prepared based on Mediouni and Dhouibi [36] with a slight modification. Briefly, 600 g wheat bran was dry heat-sterilized for 2 h at 121˚C and mixed with 250 mL auto-claved water. 120 mL corn oil was used instead of glycine, and 23 g wheat germ powder per 1000 g artificial diet was added.

## RNA extraction and cDNA synthesis

Total RNA from 10 larvae was extracted using Trizol reagent (Invitrogen) and was treated with RNase-free DNase (TaKaRa, Shinga, Japan). The absence of DNA contamination was confirmed by PCR using 18S ribosomal DNA (rDNA)-based primers (Forward: 5' CCTTT AACGAGGATCTATTGG 3' and Reverse 5' ATACTTGGCAAATGCTTTCG 3'). cDNA was synthesized by reverse transcription using a cDNA synthesis kit (Takara, Shinga, Japan).

## HSP gene cloning

Because the heat shock gene sequences in carob moth were unknown, primers were designed using alignments of gene sequences from other insects including *Chilo suppressalis*, *Bombyx mori*, *Pieris brassicae*, *Spodoptera exigua*, and *Helicoverpa armigera*. Degenerate primers were used to amplify ~550 bp gene fragments of the carob moth hsp70 and hsp90 genes, listed in Table 1.

The amplification products were resolved by gel electrophoresis, bands were excised from the gel, and a Gene All Expin Gel SV kit (GeneAll) was used to isolate the DNA. The PCR products were ligated into a pTG19-T PCR Cloning Vector (Vivantis Technologies Sdn Bhd) using T4 DNA Ligase. The plasmids were used to transform *E. coli* DH5a cells using heat shock. Colonies with inserts were screened with blue/white X-gal selection under standard ampicillin conditions. Plasmid DNA was extracted from recombinant bacterial cells using the alkaline lysis method described by Sambrook and Russell [37], and the DNA was sequenced from multiple [5–10] different independent bacterial colonies by Bioneer (Daejeon, Korea).

**Table 1. Primers used in the current study.**

| Primers | Primer forward | Primer reverse | PCR type |
|---------|----------------|----------------|----------|
| HSP70 | 5'-AACCACTTCGTTCAGGAGT-3' | 5'-TTGTTGTCCTTGGTCATGGC-3' | RT-PCR |
| HSP90 | 5'-CAGTTCGGTGTGGGTTTCTA-3' | 5'-TTGTAGAAGTCGCCGTACTCC-3' | RT-PCR |
| T7-HSP70 | 5'-TAATACGACTCACTATAG AACCACTTCGTTCAGGAGT-3' | 5'-TAATACGACTCACTATAG TTGTTGTCCTTGGTCATGGC-3' | (dsRNA synthesis) |
| T7-HSP90 | 5'-TAATACGACTCACTATAG CAGTTCGGTGTGGGTTTCTA-3' | 5'-TAATACGACTCACTATAG TTGTAGAAGTCGCCGTACTCC-3' | (dsRNA synthesis) |
| T7-GFP | 5'-TAATACGACTCACTATAG GTGGAGAGGGTGAAGG-3' | 5'-TAATACGACTCACTATAG GGGCAGATTGTGTGGAC-3' | (dsRNA synthesis) |
| re- HSP70 | 5'-AACCACTTCGTTCAGGAGT-3' | 5'-CTCCTCTGTCGCTCGGTAT-3' | RT-qPCR |
| re- HSP90 | 5'-CAGTTCGGTGTGGGTTTCTA-3' | 5'-GAGGATGACAAGCCCAAGAT-3' | RT-qPCR |
| RL32 | 5'-TGGCACCACACCTTCTAC-3' | 5'-CATGATCTGGGTCATCTTCT-3' | RT-qPCR |

## Sequence alignments and analyses

The sequences of the HSP70 and HSP90 cDNA were confirmed by a homology search of other HSP70 and HSP90 sequences known within the GenBank database of the National Center for Biotechnology Information (NCBI) website (http://www.ncbi.nlm.nih.gov/BLAST/). Phylogenetic trees were constructed by the neighbor-joining method with a Poisson correction model (1,000 bootstrap replications to check for repeatability of the results) using the MEGA 6.0 software [38].

## Preparing dsRNA

Template DNA was amplified with the gene-specific primers that possessed the T7 RNA polymerase promoter sequence at the 5' end (5-TAATACGACTCACTATAG-3) (Table 1). The PCR products were used for the preparation of dsRNA using the MEGA Script RNAi kit according to the manufacturer's instructions (Ambion, TX). The dsRNA fragments, for both HSP70-dsRNA and HSP90-dsRNA, were approximately 500 bp. The concentration and quality of dsRNAs were assessed with a Nanodrop spectrophotometer.

## Quantitative PCR (RT-qPCR)

RNA and cDNA was prepared from pools of 10 larvae for each stress and control treatment. RT-qPCR was employed to determine the expression of HSP70 and HSP90 in treated and non-treated carob moth using RealQ Plus 2x Master Mix Green (ampliqon) on a Rotor-Gene® Q system (QIA-Gene). Gene-specific primers (Table 1) were designed using Primer-Blast. A ribosomal protein gene, RL32, which is constitutively expressed in all tissues, was used as an internal control. The PCR amplifications were performed with the following cycling conditions: one cycle at 95˚C (15 min), followed by 40 cycles of denaturation at 95˚C (15 s), annealing at 55˚C for the 20s, and extension at 72˚C for 20 s. Quantitative analysis followed a comparative CT ($\Delta\Delta$CT) method [39]. A melt curve analysis was used to assess whether a single amplicon was produced in all RT-qPCR reactions. For each treatment, three replicates of 10 individuals were analyzed.

## RNA interference of HSP genes

RNAi bioassays were conducted by injecting 4 μL of the dsRNA (250 ng) solution into the hemocoel of 1-day-old fifth-instar larvae with a Hamilton microsyringe [40]. A green fluorescent protein (GFP) dsRNA (dsRNA-GFP) was used as an exogenous control (negative control). Six replicates of 80 1-day-old fifth- instar larvae were conducted for each dsRNA treatment. The survival rate after dsRNA injection was more than 90%. After 24 h, the insects were subjected to 46˚C for 100 min or -15˚C for 30 min to determine the effect of heat and cold temperatures on survival, respectively. Then mortality was assessed 24 h after the thermal treatment.

## Temperature tolerance bioassays

**High temperature.** To determine the susceptibility of the insects to high and temperatures, different developmental stages of carob moth, including egg, larval, and pupal stages were held into 9 cm Petri dishes, and adults were transferred into clear plastic cylinder bottles (3 cm diameter × 8 cm height). Insects were then exposed either to 46˚C for 1 hour using a temperature-controlled incubator with 47±1% humidity, or to -15˚C for 30 min in laboratory fridge [40]. After determination of heat and susceptibility of all developmental stages of carob moth, fifth instar larvae were exposed to different high temperatures for 100 min or 46˚C for

different exposure periods (0–120min) in order to determine the lethal high temperature and the lethal time exposure, respectively. Similarly, fifth instar larvae were exposed to different low temperatures for 100 min or -15˚C for different exposure periods (0–60 min) in order to determine the lethal low temperature and the lethal time exposure, respectively. After two-hour recovery periods at 25˚C, mortality was recorded. Individual larvae that did not move after touching with a small fine brush were counted as dead. Egg and pupal survivals were determined by hatch and adult emergence at 25˚C, respectively. The effective temperature for heat or cold tolerance induction was determined by exposing fifth-instar larvae to different high or low temperatures for 10 min and then either 46˚C for 100 min or -15˚C for 30 min. The effective exposure time for the heat tolerance induction was measured by exposing fifth-instar larvae to different periods at 37˚C and 46˚C for 100 min, while the effective exposure time for the cold tolerance induction was measured by exposing fifth- instar larvae to different periods (0, 10, 15, 20, and 30 min) at 10˚C and -15˚C for 30 min. After these temperature treatments, the survivability of individuals was determined after two-hour recovery at 25˚C [40].

## The effect of larval starvation on Heat Shock proteins (HSPs) expression and antioxidant enzymes activity

To determine the effects of starvation on HSP70 and HSP90 expressions level and antioxidant enzyme activity (SOD and catalase activities as well as malondialdehyde concentration), eighty 1-day-old fifth-instar larvae were placed into eight petri dishes (diameter of 9 cm) and were starved for 24 h and 48 h. Then, these starved larvae were used to determine SOD and Catalase activities and malondialdehyde concentration as well as HSP70 and HSP90 expressions level. Each assay was performed three times, and each assay in triplicate.

## The effect of larval Parasitism on HSPs expression and antioxidant enzyme activity

In order to study the effect of parasitism on HSP70 and HSP90 expressions level as well as on the SOD and Catalase activities and malondialdehyde concentration, eighty 1-day-old fifth-instar larvae were exposed to a pair of *Habroacon hebetor* (Hymenoptera, Braconidae), were obtained from University of Tehran Biological Control Lab, for a 24 h period (a honey solution of 10% v/v was provided as a food source). After 24 h, parasitized larvae (paralyzed larvae) were collected and used to determine SOD and catalase activities and malondialdehyde concentration. Parasitized larvae were also used to assess the HSP70 and HSP90 expressions level (see Quantitave PCR section above). Each enzyme assay was performed three times, and each assay carried out in triplicate.

## Sample preparation

The fifth-instar (L5) larvae from control and treatments were collected and were transferred in an ice-cold saline solution (NaCl, 10 mM) and homogenized by a glass pestle. The homogenates were centrifuged at 10000 rpm for 15 min at 4˚C. The supernatants were stored at –20˚C for subsequent analyses.

## Antioxidant assays

SOD activity was measured using the procedure described by Beauchamp and Fridovich [41]. Briefly 0.1 ml enzymatic extract was added into 1.5 ml 0.05 M PBS (Phosphate buffer saline, pH 7.8), 0.3 ml 0.1 mM EDTA (Ethylenediaminetetraacetic acid), 0.3 ml 0.13 M methionine, 0.3 ml 0.75 mM NBT (Nitrotetrazolium Blue chloride), 0.3 ml 0.02 mM riboflavin. Incubation

was initiated with exposure to fluorescent light (4,000 lux) for 10 min, and the change in absorbance was recorded at 560 nm. Catalase activity was measured using the method of Wang et al. [10] in which 50 μL of sample and 500 μL of hydrogen peroxide (1%) were incubated at 28˚C for 10 min before determining changes in optical density at 240 nm.

## Malondialdehyde concentration

A method described by Bar-Or et al. [42] was used to determine the concentration of malondialdehyde (MDA) in the control and treated individuals. 100 μL of 20% trichloroacetic acid and 50 μL of the sample was mixed and centrifuged at 15,000g for 10 min at 4˚C. Then, the supernatant was mixed with 100 μL of 0.8% TBA reagent and re-incubated at 100˚C for 60 min before reading absorbance at 535 nm. The MDA concentration is reported as the amount of MDA produced per mg protein using a molar extinction coefficient of $1.56 \times 10^5$ $M^{-1}$ $cm^{-1}$.

## Protein assay

Protein content was determined by the Bradford [43] and using bovine serum albumin as the standard protein.

## Statistical analysis

Means were compared by the least squared difference (LSD) tests of one-way ANOVA using SPSS software. Values were considered statistically different if $p < 0.05$. Drawing graphs were conducted using Excel 2013.

# Results

## Heat shock genes identification

The HSP70 and HSP90 gene fragments from *E. ceratoniae* were PCR-amplified from cDNA using primers that were designed to anneal to conserved regions of other insect HSP70 and HSP90 genes. Analysis of the carob moth putative HSP70 gene fragment sequence confirmed that it was indeed similar to other insects' HSP70 genes, showing highest identity (93%) to that of *Cadra cautella* (almond moth). A phylogenetic analysis of the putative HSP70 gene sequence of the carob moth and other insects likewise confirmed that the carob moth's gene is most similar to HSP70 genes of other lepidopterans, and in particular, to moths of the family Pyralidae (Fig 1A) This cDNA sequence has been deposited in GenBank under accession number MN529255. A phylogenetic analysis of HSP90 genes from several lepidopteran insects likewise confirmed that the carob moth gene's sequence is most similar to that of another lepidopteran species, *Bicyclus anynana* (squinting bush brown moth; Fig 1B). This cDNA sequence has been deposited in GenBank under accession number MN529256.

## Temperature tolerance bioassays

Susceptibility of the insects to high (S1A Fig) and low (S1B Fig) temperatures was initially assessed for all developmental stages by exposing the insects to 46˚C for one hour or -15˚C for 30 min, respectively. Temperatures were chosen based on literature review that shows these points usually used for Pyralid moth control in the storehouses [40]. Also, these temperature and exposure lengths were selected to facilitate comparisons with similar studies. Larval mortality was recorded after two-hour recovery periods at 25˚C. Egg and pupal survivals were determined by hatch and adult emergence at 25˚C, respectively. Late instar larvae (i.e., fourth and fifth instars) were the most tolerant of the high-temperature treatment, with L4 and L5 survival rates of approximately 50 and 60%, respectively (S1A Fig). In contrast, eggs, L1, L2,

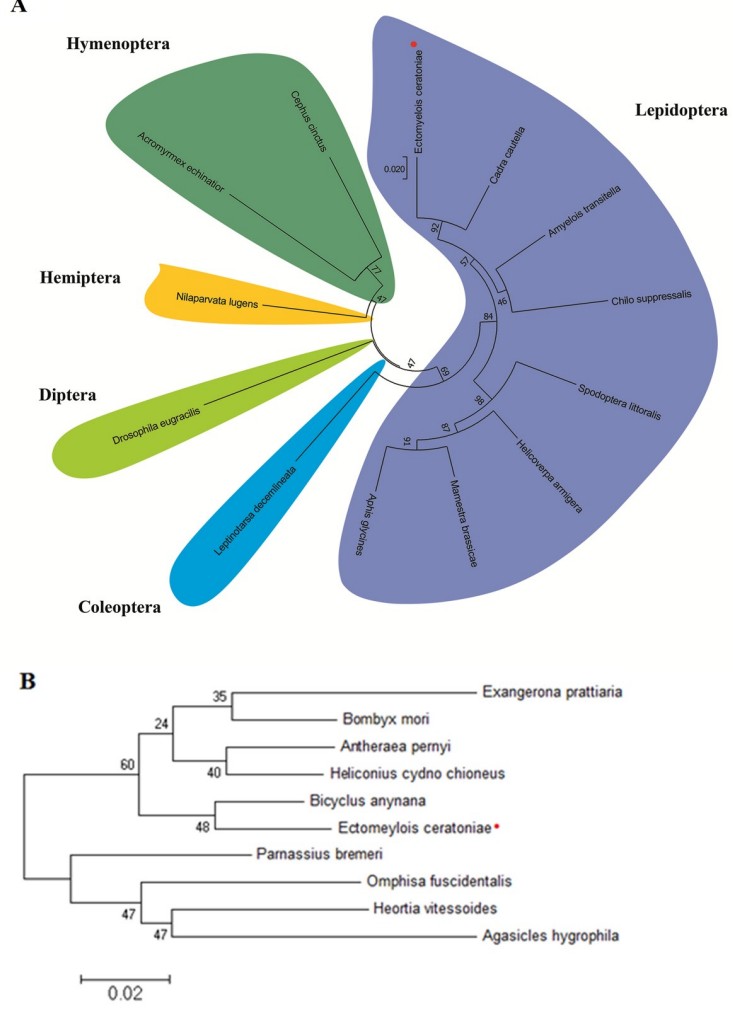

**Fig 1. Phylogenetic relationship of sequences of HSP70 and HSP90 of carob moth and other insects.** (A) An unrooted neighbor-joining tree of HSP70 cDNA sequences was constructed with bootstrap support based on 10,000 replications shown at the relevant branch points. *Cadra cautella* (MF067297.1), *Amyelois transitella* (XM_013331933.1), *Mamestra brassicae* (AB251896.1), *Helicoverpa armigera* (XM_021342205.1), *Chilo suppressalis* (GU726137.1), *Spodoptera littoralis* (KC787696.1), *Leptinotarsa decemlineata* (KC544268.1), *Drosophila eugracilis* (XM_017212762.1), *Nilaparvata lugens* (KX976471.1), *Acromyrmex echinatior* (XM_011062182.1), *Cephus cinctus* (XM_015747637.2). (B) Neighbor-joining tree of predicted protein sequences of HSP90 from the carob moth and other lepidopteran insects. *Heortia vitessoides* (AWX67685.1), *Exangerona prattiaria* (ADM26739.1), *Antheraea pernyi* (APX61060.1), *Parnassius bremeri* (APP93215.1), *Agasicles hygrophila* (ALP48323.1), *Bombyx mori* (AEB39782.1), *Omphisa fuscidentalis* (ABP93404.1), *Bicyclus anynana* (AFM73651.1), *Heliconius cydno* chioneus (AFQ40718.1).

pupa, and adults, were all equally adversely affected, suffering >96% mortality rate with this high temperature treatment. (S1A Fig).

Regarding cold tolerance, L5 larvae were the most tolerant of the low-temperature treatment, with only 20% of the insects surviving the -15˚C treatment (S1B Fig). Again the same trend as high temperature survivorship was observed for all other insect developmental stages i.e. >96% of eggs, L1, L2, pupa, and adults died from this cold treatment (S1B Fig). Thus, for both low temperature and high temperature treatments, late larval stages were the most tolerant.

Since fifth instar larvae were the most tolerant of both heat and cold treatments, they were chosen for subsequent treatments. All L5 larvae could be killed by subjecting them to 46°C for 120 min, or to -15°C for 50 min (S2 Fig). Mortality was time-dependent with both the high or low treatments, with median lethal times ($LT_{50}$) of 67.5 min (95% CI: 65.0–70.0°C) and 16.6 min (95% CI: 14.3–18.7°C) for heat and cold treatments, respectively. Further heat and cold treatments at different times and temperatures were tested, and not surprisingly, demonstrated that 100% mortality could be achieved at higher or lower temperature treatments in less time (S3 Fig).

Pre-exposure of fifth instar larvae to different sublethal high and low temperatures significantly increased survival rates; the pre-exposures at 37°C or 10°C for 10 min resulted in the highest survival rates of L5 larvae following their subsequent exposure to the lethal high (46°C for 100 min) and low (-15°C for 30 min) temperature treatments, respectively (S4 Fig). Based on these results, the insects were then subjected to these two pretreatment temperatures for variable durations and observed that heat or cold tolerances could be induced within as little as 10 min pretreatment (S5 Fig). The highest survival rates were observed in insects that were pretreated to 37°C for 20 min or 10°C for 30 min, with survival rates of 56.7 ± 2.2 and 31.7 ±1.2, respectively.

## Changes in HSP transcript levels during stress

Quantitative RT-PCR confirmed that both of the newly-identified HSP genes of the carob moth were induced by either heat or cold shock. Exposure to a moderately high temperature significantly increased the expression level of HSP70 and HSP90 by 3.0 and 3.5-fold in comparison with the controls, respectively (Fig 2). Also, pretreatment with moderately cold temperature similarly increased the expression levels of HSP70 and HSP90 by 4.0 and 2.3-fold compared to the controls, respectively (Fig 2). In response to high-temperature treatments, HSP90 transcript levels increased to a greater degree than did the transcript levels of HSP70, while at the cold temperature, HSP70 was induced more than that of HSP90.

HSP90 transcript levels also increased significantly when larvae were subjected to starvation; HSP90 transcripts increased 1.6-fold and 2.1-fold in starved larvae compared to control

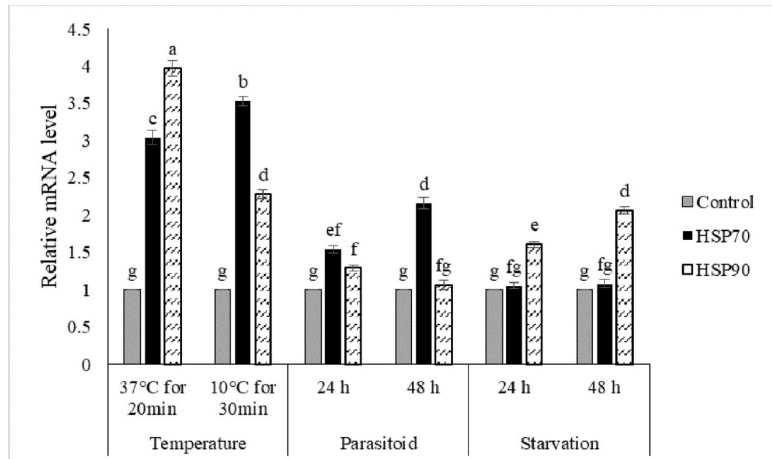

**Fig 2. Level of HSP70 and HSP90 transcripts following a moderate heat (37°C for 20 min) and cold (10°C for 30 min) treatment, relative to their respective levels at normal rearing temperature (26°C), parasitism by *H. hebetor* at 24 h and 48 h post parasitism, and at 24h and 48h post-starvation.** The values represent the means and standard errors for three replicates of 10 individuals. Different letters denote significantly different values from one another (LSD).

after 24 h and 48 h starvation, respectively. In contrast, HSP70 transcripts were not significantly affected by starvation relative to the non-starved controls (Fig 2). In wasp-parasitized larvae, HSP70 transcripts increased 1.5 and 2.2-fold after 24 h and 48 h post-parasitism, respectively, relative to non-parasitized larvae (Fig 2). HSP90 transcripts were not significantly affected by the parasitism.

## RNAi-mediated knockdown of HSP transcripts

Injection of dsHSP70 and dsHSP90 into fifth instar larvae suppressed the expression level of each of these targeted genes. The HSP70 mRNA level was significantly reduced by 9.5 and 4.0-fold at 24 h and 48 h post dsRNA injections compared to controls, respectively (Fig 3A). HSP90 expression level was not as strongly reduced as those of HSP70, but it was significantly reduced 1.4-fold relative to the controls 24 h post dsRNA injections (Fig 3B). This transcript reduction did not persist, as HSP90 transcript levels returned to normal by 48 h post-injection.

To determine the role of these HSPs in heat and cold tolerance induction, L5 larvae were injected with each gene-specific dsRNA. The mortality rates significantly increased by 2.6 and 3.2-fold in dsHSP70- and dsHSP90- treated larvae in comparison with controls in heat treatments (Fig 4A). In cold treatments, injection of dsHSP70 and dsHSP90 resulted in increased mortality rates by 2.1 and 1.5-fold in comparison with the controls, respectively (Fig 4B). While increases of HSP70 and HSP90 transcript levels coincided with enhanced thermotolerance to both heat and cold stresses (Fig 2), the knockdown of HSP70 and HSP90 transcripts resulted in greater mortalities, which provides good supporting evidence that these genes' expression are associated with improved protection from heat and cold temperature stresses (Fig 4).

## Oxidative stress responses

Evidence of oxidative stress was observed in all stress treatments by measuring MDA. MDA concentrations in larvae treated with moderate low (10°C for 30 min) or moderate high temperatures (37°C for 20 min) increased 2.3 and 1.7-fold of that of controls, respectively (Fig 5A). Envenomation of carob moth fifth-instar larvae by *H. hebetor* led to a significant increase in MDA concentrations; the MDA concentration in parasitized larvae was 1.3-fold higher than that of control (unparasitized larvae) (Fig 5A). In response to starvation, MDA concentrations increased only slightly (1.1-fold) by 24 h, but had increased significantly (1.9-fold) by 48 h post-starvation (Fig 5A).

SOD activity significantly increased when larvae were exposed to moderate low or high temperatures (Fig 5B). SOD activity in treated larvae increased 1.4-fold for moderate high temperature (294.51 μm.min$^{-1}$.mg protein$^{-1}$) and 1.3-fold for moderate low temperature (277.67 μm.min$^{-1}$.mg protein$^{-1}$) compared to controls. Envenomation by *H. hebetor* also affected SOD activity (Fig 5B), increasing 1.4-fold compared to controls (Fig 5B). Starvation also induced SOD activity of carob moth larvae, increasing 1.3 and 1.6-fold at 24h and 48h after starvation in comparison with controls, respectively (Fig 5B).

Treating fifth instar carob moth larvae with moderate low or high temperatures statistically increased the CAT activity compared to controls. Exposure to 37°C for 20 min or 10°C for 30 min increased CAT activity 4.7 and 2.6-fold compared to controls, respectively (Fig 5C). Parasitism by the parasitoid wasp significantly increased CAT activity 1.7-fold 24 hours post-attack (Fig 5C). CAT activity increased in response to starvation 2.6 and 3.7-fold after 24 h and 48 h starvation, respectively(Fig 5C).

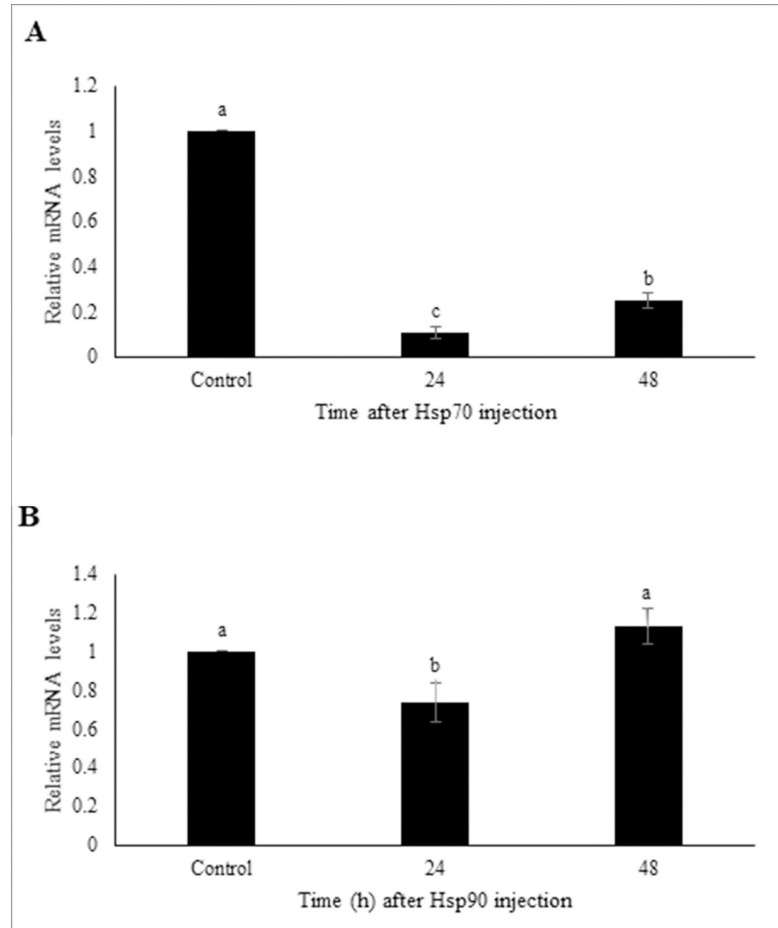

**Fig 3. Extent of transcript knockdown following injection of L5 larvae with HSP70, HSP90, or GFP (negative control) dsRNAs.** Following dsRNA injection, insects were allowed to recover at 24 or 48 h days before being subjected to RNA extraction and qRT-PCR analysis. A) Level of HSP70 transcripts following HSP70-dsRNA injections relative to GFP-dsRNA injected controls. B) Level of HSP70 transcripts following HSP70-dsRNA injections relative to GFP-dsRNA injected controls. Transcript levels were standardized relative to rl32 transcripts levels. The values represent the means and standard errors for three replicates of 10 individuals. Different letters denote significantly different values from one another (LSD).

## Discussion

The carob moth is a serious pest of a variety of stored foods, causing direct or indirect damage to these products. To avoid using chemical insecticides on our foods, alternative methods such as heat or cold treatments could potentially be used to control pest insects of our stored products. The temperature and duration of treatments needed to control insect pests will be dependent on the species and developmental stage of the insects to be treated. In this study, the susceptibility of all developmental stages of carob moth to high and low temperatures was investigated. Heat treatments of 46°C for 100 min or cold treatments of -15°C for 30 min caused significant mortality of most developmental stages of this pest insect. However, the late larval instars were the most tolerant to these temperatures, with more extreme temperatures or longer durations of temperature stress required to kill these more robust stages. Similar temperature treatments have been reported to be lethal to the other lepidopteran insects. Wang et al. [44] reported that exposure to 47°C or -7°C for two hours killed all larval instars of the noctuid moth *Xestia c-nigrum*. Likewise, exposures to 44°C for 70 min were sufficient to kill

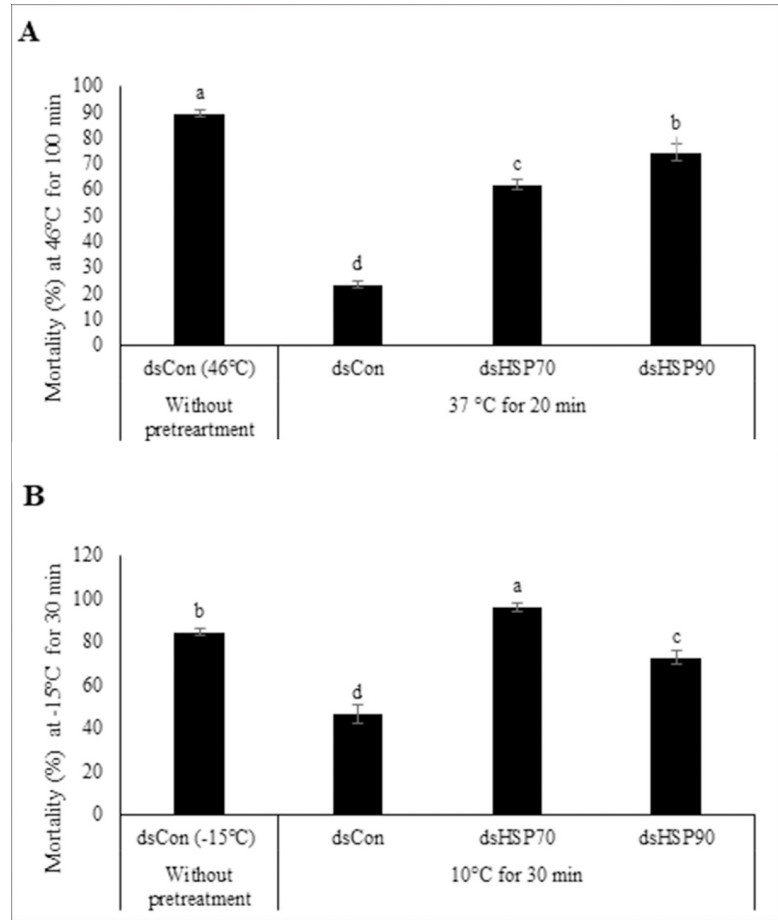

**Fig 4.** Effect of RNAi-mediated HSP genes silencing transcript knockdown on mortality rates in carob moth larvae subjected to (A) a moderate heat (37˚C for 20 min) or (B) a moderate cold (10˚C for 30 min) pretreatment, followed by exposure to a higher temperature treatment. Insects were injected with dsRNA targeting HSP70, HSP90, or GFP (negative control) genes' transcripts, allowed to recover for 24 h and then subjected to the temperature (pre) treatments. The values represent the means and standard errors for percent survival for the replicates of 80 individuals. Different letters denote significantly different values from one another (LSD).

most developmental stages of the Indian meal moth (*Plodia interpunctella*), but like the carob moth, the late larval instars showed greater tolerance of the extreme temperatures [40]. The reasons for the greater tolerance to extreme temperatures of late instar larvae have not been fully explored, but in this current study, the production of protective molecules or increased expression of existing protective molecules were examined to help provide some insights into the adaptive systems in carob moth larvae.

All four stresses examined in this study induced oxidative stress responses in the carob moth larvae. MDA levels increased following exposure to heat, cold, starvation and parasitism, indicating evidence of lipid peroxidation. The insects responded to this stress by increasing both SOD and CAT to manage the increased levels of ROS compounds. In conjunction with these curative antioxidant responses, the stress treatments also increased expression of one or both of the two HSPs examined in this study. HSPs act as molecular chaperones to stabilize the structure of proteins in cells subjected to many stresses. Increased transcription of HSP genes has been often observed in insects subjected to a diversity of stresses, including heat, cold, starvation and parasitism (reviewed in 1). In this study, pre-exposure of the carob moth larvae to

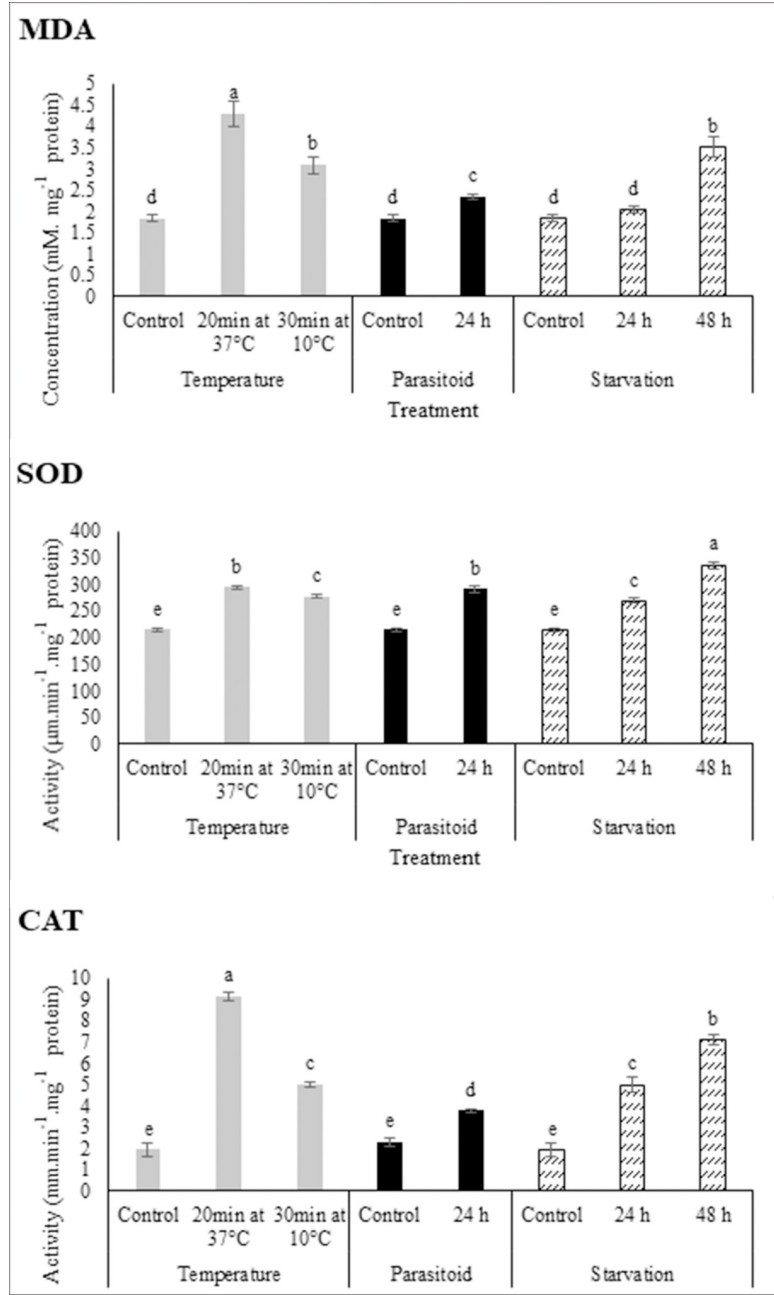

**Fig 5. The effect of pre-exposer to moderate heat (37˚C for 20 min) and cold (10˚C for 30 min) temperatures, parasitism by *H. hebetor* at 24 h and 48 h post parasitism, and 24h and 48h post starvation on antioxidant system including SOD and CAT activity and MDA concentration in carob moth fifth instar larvae.** The values represent the means and standard errors for three replicates of 10 individuals. Different letters denote significantly different values from one another (LSD).

sublethal hot or cold temperatures enhanced their survival at the more extreme temperatures. QRT-PCR confirmed that these moderate heat and cold pre-treatments of carob moth larvae increased the expression of both HSP70 and HSP90 genes. Increased HSP transcription rates in pretreated insects have frequently been associated with increased thermotolerance (reviewed in 1). RNAi-mediated knockdown of either HSP70 or HSP90 transcripts in the

carob moth larvae greatly reduced the ability of the insects to withstand extreme temperature treatments, which provides compelling evidence that these genes' proteins are indeed critical for enhanced thermotolerance. The utility of RNAi as a molecular biology tool to provide evidence of gene function is well-recognized in many eukaryotic species, but RNAi has not always been effective in some lepidopteran insects [45]. In the carob moth, injection of dsRNA into the hemocoel proved effective in reducing the targeted genes' transcripts, demonstrating that this lepidopteran species is somewhat amenable to this useful molecular biology tool. It is worth noting however, knockdown of HSP90 (25% at 24 h post-injection) was considerably weaker than that of HSP70 (90% at 24 h post-injection), and the knockdown was only temporary for HSP90. Similarly weak and transient RNAi-mediated knockdowns are common in many lepidopteran species [45, 46], but in the carob moth, the knockdown was nevertheless sufficient to confirm that HSP90 plays an important role in conferring thermotolerance.

Interestingly, the extent of induction differed somewhat between the two HSPs with the different temperature treatments. While both HSPs' transcripts increased with heat and cold, exposure to high temperature caused a greater proportional increase in the expression of HSP90 relative to HSP70, whereas exposure to a low temperature caused a greater induction of HSP70 relative to HSP90. The differential increases of the two HSPs' transcripts with variable stresses reflects differences in both their inducibility with different stresses as well as their constitutive levels of expression prior to a stress. While both HSPs are capable of acting as molecular chaperones to assist in protein folding under non-stressed conditions, they do have different roles that may reflect different induction rates with different stresses. HSP70 primarily assists in folding of newly translated proteins, while HSP90 is thought to assist with maintaining or refolding pre-existing proteins, particularly those with hydrophobic substrate clefts. They also have divergent roles in determining the fate of proteins, with HSP70 facilitating ubiquitination and proteasome-directed degradation, while HSP90 can inhibit ubiquitination [47]. In this study, differential induction of HSP70 and HSP90 was also seen in response to starvation and parasitism. HSP70 transcription did not increase significantly in response to starvation, whereas HSP90 was up-regulated sharply with this stress. Starvation is a stressor that affects water, ion balance, and energy availability [14]. With these imbalances, new protein synthesis may slow down, and hence, perhaps less HSP70 is required, while more HSP90 may be required to deal with the perturbations of the cellular proteins during starvation stress. Differential HSP responses have been observed in other insects subjected to starvation. Sinclair et al. [20] observed that in *D. melanogaster*, HSP70 was induced by cold but not by starvation, while Tian et al. [35] found that starvation in house fly (*Musca domestica*) larvae induced increased expression of only HSP27, while other HSPs examined were not affected.

In contrast, parasitism by *H. hebetor* in carob moth larvae enhanced HSP70 transcription, while HSP90 transcript levels were unchanged. Similar observations were observed by Shim et al. [23], where parasitized larvae of *Plodia interpunctella* by *H. hebetor* enhanced HSP70 but not HSP90 transcript levels. The differential response of these HSPs to this particular stressor may reflect the host's heightened immune responses, and it will be of particular interest to explore other infectious agents in this insect to determine if they also elicit differential HSP responses. However, parasitism is a particularly complex stressor, as there could be considerable interplay between host and parasite. Like each of the other stressors, parasitism also resulted in increased ROS levels and increased antioxidant enzymes levels. With this particular stress however, the changes in antioxidant activity cannot be exclusively attributed to just the host, as the parasite could also be a source of both ROS and antioxidant activity, to counter the host insect's defenses [48].

Examining insects' responses to different stresses has provided us with excellent insights into how many stresses can induce complex defense systems, some of which are shared across

species, and some of which are differentially regulated in different species or in response to different stressors. Understanding insects' tolerances to stress may also help guide us in the development of new, environmentally safer approaches to insect control, reducing our reliance on our currently available chemical insecticides. However, given the diversity of stress responses that insects have, we may find that using abiotic or biotic stress control methods may need to be adapted to each pest for the most effective control methods.

## Supporting information

**S1 Fig. Susceptibility of different developmental stages to extreme temperatures.** Insects were subjected to (A) heat: 46˚C for one hour or (B) cold: -15˚C for 30 min, and the percent dead was assessed.
(PDF)

**S2 Fig.** Effect of exposure times on mortality of carob moth L5 larvae using (A) heat: 46˚C for 120 min or (B) cold: -15˚C for 30 min.
(PDF)

**S3 Fig.** Effect of different (A) high or (B) low temperatures on mortality of carob moth L5 larvae.
(PDF)

**S4 Fig. Induction of extreme temperature tolerance in carob moth L5 larvae.** Insects were pretreated for 10 min at various temperatures and then subjected to (A) 46˚C for 100 min or (B) -15˚C for 30 min.
(PDF)

**S5 Fig.** Induction of extreme temperature tolerance by varying duration of (A) 37˚C pretreatments or (B) 10˚C pretreatments and then subjected to (A) 46˚C for 100 min or (B) -15˚C for 30 min.
(PDF)

## Acknowledgments

The authors thank the University of Tehran for the provision of necessary facilities to carry out experiments. This research was supported by funds to Ali R. Bandani from the University of Tehran, funds to Steven Whyard from Natural Sciences and Engineering Research Council, and a scholarship from Iran's Minister of Science Research and Technology to Saeed Farahani.

## Author Contributions

**Conceptualization:** Ali R. Bandani, Steven Whyard.

**Data curation:** Saeed Farahani.

**Funding acquisition:** Ali R. Bandani, Steven Whyard.

**Investigation:** Saeed Farahani.

**Methodology:** Seyed Hossein Goldansaz.

**Resources:** Houshang Alizadeh.

**Software:** Houshang Alizadeh.

**Supervision:** Ali R. Bandani.

**Writing – original draft:** Saeed Farahani.

**Writing – review & editing:** Ali R. Bandani, Houshang Alizadeh, Steven Whyard.

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
