## [Decision Letter · Decision Letter 0]

9 Oct 2019

PONE-D-19-21471

Differential expression of heat shock proteins and antioxidant enzymes in response to temperature, starvation, and parasitism in the Carob moth larvae, Ectomyelois ceratoniae (Lepidoptera: Pyralidae)

PLOS ONE

Dear Dr Bandani,

Thank you for submitting your manuscript to PLOS ONE. After careful consideration, we feel that it has merit but does not fully meet PLOS ONE’s publication criteria as it currently stands. Therefore, we invite you to submit a revised version of the manuscript that addresses the points raised during the review process.

Please pay close attention to the requests of the referees. Referee 2 is concerned about the lack of important citations.

We would appreciate receiving your revised manuscript by Nov 23 2019 11:59PM. To enhance the reproducibility of your results, we recommend that if applicable you deposit your laboratory protocols in protocols.io, where a protocol can be assigned its own identifier (DOI) such that it can be cited independently in the future. For instructions see: http://journals.plos.org/plosone/s/submission-guidelines#loc-laboratory-protocols

We look forward to receiving your revised manuscript.

Kind regards,

Marcelo Hermes-Lima, PhD

Academic Editor

PLOS ONE

Journal Requirements:

2. In your Methods section, please provide additional location information of the collection site of Ectomyelois ceratoniae, including geographic coordinates for the data set if available."

4. In your Methods section, please provide additional details regarding the Habrobracon hebetor used in your study and ensure you have described the source. For more information regarding PLOS' policy on materials sharing and reporting, see https://journals.plos.org/plosone/s/materials-and-software-sharing#loc-sharing-materials.

5.  We suggest you thoroughly copyedit your manuscript for language usage, spelling, and grammar. If you do not know anyone who can help you do this, you may wish to consider employing a professional scientific editing service. 

Reviewers' comments:

Reviewer's Responses to Questions

**Comments to the Author**

1. Is the manuscript technically sound, and do the data support the conclusions?

Reviewer #1: Yes

Reviewer #2: Yes

2. Has the statistical analysis been performed appropriately and rigorously? 

Reviewer #1: Yes

Reviewer #2: I Don't Know

3. Have the authors made all data underlying the findings in their manuscript fully available?

Reviewer #1: Yes

Reviewer #2: Yes

4. Is the manuscript presented in an intelligible fashion and written in standard English?

Reviewer #1: Yes

Reviewer #2: Yes

5. Review Comments to the Author

Reviewer #1: In this study, Bandani and colleagues set out to investigate the expression of heat shock proteins, the activities of two antioxidant enzymes and levels of lipid peroxidation in Ectomyelois ceratoniae, exposed to heat, cold, starvation, and parasitism stresses. First, authors successfully cloned and sequenced fragment sequences of E. ceratoniae HSP70 and HSP90 mRNA, this allowed them to design specific primers to assess the transcript levels for these proteins and to perform RNA interference experiments. Noteworthy, animals experiments began with the assessment of the animal’s tolerance to temperature stresses depending on its developmental stage; this supported the design of the following experiments. Authors should note that although L5 were the most resistant to temperature stress, this could not be the case for the other stresses (starvation and parasitism). Temperature stress induced the expression of HSPs at the mRNA levels, and RNA interference significantly affected the survivability. Interestingly, parasitism and starvation induced different HSPs. All stress protocols (heat, cold, parasitism, and starvation) elicited lipid peroxidation and increased the activities of SOD and catalase. These results were expected, since the activation of antioxidant defenses and elevated chaperone proteins are well-known components of the general stress response (https://www.ncbi.nlm.nih.gov/pubmed/15709958). The Discussion provides a reasonable interpretation of the results in the framework of previous studies. Limitations are mentioned, new hypotheses are defined, and a foundation for future studies is provided. I have no further recommendations, except for some minor numbered comments below:

1. Line 39: I think the abstract could benefit from a concluding take home message at the end, instead of ending with a description of the results of starvation and parasitism experiments.

2. Line 61: Consider including oxygen deprivation in the list of stresses that induce the antioxidant system of insects (see https://www.ncbi.nlm.nih.gov/pubmed/26851497/
https://www.ncbi.nlm.nih.gov/pubmed/26408245 ).

3. Line 106-107: Did you mean 18S ribosomal DNA (rDNA)-based primers? Target RNA would not give you information about DNA contamination.

4. Line 124: It would be useful to briefly mention the methods used for sequencing HSPs fragments.

5. Line 161: Please briefly describe how mortality/survivability was assessed, especially for eggs. Was it based on coloration? Movement? Please clarify.

6. Line 201: Consider renaming this section, since MDA is not an antioxidant.

7. Line 203: Did the authors use any protease inhibitor during the homogenization for enzyme assays? The use of single protease inhibitors (i.e. phenylmethylsulfonyl fluoride) or commercial cocktails is a common procedure during tissue homogenization prior to the measurement of enzyme activities. This procedure is applied to maintain and preserve protein functionality after tissue homogenization.

8. Line 215: Did authors use any antioxidant (e.g., BHT) to avoid lipid peroxidation during the analytical phase of the assay?

9. Line 221: Conventionally, author should mention p value considered to indicate significant differences.

10. Line 234: Italicize Bicyclus anynana

11. Line 276: I think “suboptimal” is not the best term here. Sublethal?

12. Figure 5: I was a little confused in the 25-degree pre-treatment for 10 minutes. How was this performed if animals were maintained at “25 +/- 1 °C”? If animals were maintained at 26°C how a ten-minute pre-treatment could take place?

13. Figures 5 and 6: If available, plot data for larvae not subjected to any pre-treatment in figures 5 and 6 for comparative purposes.

14. Figures: Considering that there are ten figures, I would suggest authors to place most of the survivability data in supplemental material. Do not get me wrong, these results are fundamental for the further experiments and it is a huge positive point of this study (not all studies justify their experimental design (dosage, time of exposure, developmental stage, etc) based on actual data. But I think some of the data might be move to supplemental material (keeping the description in the main text) to make the manuscript more objective.

15. Figure 7: The meaning of the different columns fills was not defined, include legends for the column patterns in the figure and/or explain them in the legend.

16. Figures: For clarity, consider placing panel labels (A, B, etc) on the upper-left position, not centered.

Reviewer #2: The goal of this MS is to understand the role of AOEs and HSPs in the Carob moth as it relates to temperature (low and high), starvation, and parasitism. The authors use a robust strategy that involves gene expression, biochemistry, survivorship and loss of function experiments. I think the results are interesting and worthy of publication.

My main concern with the MS has to do with background work and literature. The authors consistently cite new literature, in multiple cases papers that are less than five years old when older examples (10 to 15 yrs old) provide better background information and data. Specifically, the entire literate on heat treatments use in commodities handling is ignored; a growing area of research as the authors know. They also ignore, probably indirectly, a lot of the classic insect HSP and AOE work by Lee, Denlinger, and others. I was surprised and glad to see references to Sinclair’s HSP work but most of the author’s background comes from recent studies that are not very thorough. Additionally, the literature on temperature pre-treatments is also quite vast. It includes work on low temperature pretreatments like rapid cold hardening (also of Lee and Denlinger), rapid heat hardening, and hormesis. I mention this not to point out a flaw, but to suggest to the authors that the literature not being presented in here, might provide answers and mechanistic explanations that would strengthen their MS.

My second main concern has to do with the tone of the MS. This is an insect MS written for an insect audience but sent to a general broad journal. The authors never explain the reason for their treatments. Why starvation? Why are those temperatures chosen? Why the lengths of treatment? And the parasitism? Does biological control of Ectomyelois involve parasitoids? The authors need to explain a lot about the reason behind their treatments and the biology of the moth in a non-entomology journal. I am an entomologist and work in a similar area, and I do not understand the point of starvation, in this context. Without explanations as to why certain treatments were chosen, the MS reads like a bunch of random experiments pasted together and I suspect that is not the case and that the treatments were chosen rather carefully. If this is a paper about “aoes and hsps in response to common non-pesticide control tactics for Carob moth” then fine, but please do tell us.

I have more specific comments below:

Line 53, the appropriate reference about insect antioxidant systems is not paper from 2015. I believe there are multiple papers in the 1990s (archives of insect biochemistry had an entire antioxidant issue in 1997, I believe), and dozens of other papers between then and 2015.

Line 57, what is the reference for all this insect non-enzymatic antioxidant explanation?

Line 60, the authors mention the different stresses that antioxidant/hsps have been involved with in insects. They exclude radiation (UV, gamma, X-ray), dehydration, anoxia, freezing, and wasp venom; which is relevant to their own point.

Line 64, HSP are involved in various cellular stressors. It would be a good idea to list them and provide references for each.

Line 65, the accepted short hand for small HSPs is smHSPs not sHSPs.

Line 162, it is here that the authors should explain the reason behind these temperatures and exposure times.

Line 177, since the authors do not reference any literature in rapid cold or rapid heat hardening, they should explain why they are not using standard temperatures and times to induce their rapid hardening protection. It would also be a good idea to cite some of that literature as background.

Line 190, the replication strategy is not clear. It reads, “performed three times, and each assay in triplicate.” But it also says 80 larvae in 8 petri dishes. Does that mean 9 sets of 80 larvae total per treatment (3 times with 3 each)?

Line 253, unless I missed it, I cannot find when mortality was counted. That information should also be here in the results. Was it at 24, 48, or xx hrs?

Line 344, responses is misspelled.

Line 360, it should be Fig 10B not 6B, right?

Line 394, “the insects responded accordingly by increasing both SOD and CAT…” They did, however lipid peroxidation damage was detected anyway. Is this really a protective release or a curative one? The authors make it sound like the insects are protecting against oxidative damage, but ox damage already happened. The statement needs revision.

Line 410, Basically Carob moth is good for RNAi. Is it? The hsp70 knockdown was good but the 90 out was a weak knockdown that barely lasted a day. It was not very persistent. This is a clear difference that should be addressed in the MS. I would argue that RNAi needs to be evaluated in this species and perhaps that big JIP 2011 paper has some ideas that could help in this system.

Line 417, hsps “both shared and different mechanisms”. This conclusion type statement is problematic. Hsps 70 and 90 work very differently. They are activated very differently because they rely on different molecules (ATP vs Ca). Their roles are different and while hsp70 is universally present in most stress, 90 is not due to the way it works and not some unknown mechanism. This statement needs revision.

6. PLOS authors have the option to publish the peer review history of their article (what does this mean?). If published, this will include your full peer review and any attached files.

Reviewer #1: No

Reviewer #2: No

---

## [Author Response · Author response to Decision Letter 0]

12 Dec 2019

Please see attached "Response to Reviewers" document.

---

## [Decision Letter · Decision Letter 1]

8 Jan 2020

Differential expression of heat shock proteins and antioxidant enzymes in response to temperature, starvation, and parasitism in the Carob moth larvae, Ectomyelois ceratoniae (Lepidoptera: Pyralidae).

PONE-D-19-21471R1

Dear Dr. Bandani,

We are pleased to inform you that your manuscript has been judged scientifically suitable for publication and will be formally accepted for publication once it complies with all outstanding technical requirements.

With kind regards,

Marcelo Hermes-Lima, PhD

Academic Editor

PLOS ONE

Additional Editor Comments (optional):

Reviewers' comments:

Reviewer's Responses to Questions

**Comments to the Author**

1. If the authors have adequately addressed your comments raised in a previous round of review and you feel that this manuscript is now acceptable for publication, you may indicate that here to bypass the “Comments to the Author” section, enter your conflict of interest statement in the “Confidential to Editor” section, and submit your "Accept" recommendation.

Reviewer #1: All comments have been addressed

Reviewer #2: All comments have been addressed

2. Is the manuscript technically sound, and do the data support the conclusions?

Reviewer #1: Yes

Reviewer #2: Yes

3. Has the statistical analysis been performed appropriately and rigorously? 

Reviewer #1: Yes

Reviewer #2: Yes

4. Have the authors made all data underlying the findings in their manuscript fully available?

Reviewer #1: Yes

Reviewer #2: Yes

5. Is the manuscript presented in an intelligible fashion and written in standard English?

Reviewer #1: Yes

Reviewer #2: Yes

6. Review Comments to the Author

Reviewer #1: (No Response)

Reviewer #2: The MS has been dramatically improved in a way that adds scientific soundness and provides the appropriate background information where required. The authors did a commendable job of addressing the concerns.

7. PLOS authors have the option to publish the peer review history of their article (what does this mean?). If published, this will include your full peer review and any attached files.

Reviewer #1: No

Reviewer #2: No

---

## [Editor Report · Acceptance letter]

13 Jan 2020

PONE-D-19-21471R1 

Differential expression of heat shock proteins and antioxidant enzymes in response to temperature, starvation, and parasitism in the Carob moth larvae, *Ectomyelois ceratoniae* (Lepidoptera: Pyralidae) 

Dear Dr. Bandani:

I am pleased to inform you that your manuscript has been deemed suitable for publication in PLOS ONE. Congratulations! Your manuscript is now with our production department. 

With kind regards,

on behalf of

Dr. Marcelo Hermes-Lima 

Academic Editor

PLOS ONE